# The Fractional Preferential Attachment Scale-Free Network Model

**DOI:** 10.3390/e22050509

**Published:** 2020-04-29

**Authors:** Rafał Rak, Ewa Rak

**Affiliations:** 1College of Natural Sciences, Institute of Physics, University of Rzeszów, Pigonia 1, 35-310 Rzeszów, Poland; 2College of Natural Sciences, Institute of Mathematics, University of Rzeszów, Pigonia 1, 35-310 Rzeszów, Poland; ewarak@ur.edu.pl

**Keywords:** complex networks, scale-free networks, fractal networks, models of complex networks, universality

## Abstract

Many networks generated by nature have two generic properties: they are formed in the process of preferential attachment and they are scale-free. Considering these features, by interfering with mechanism of the preferential attachment, we propose a generalisation of the Barabási–Albert model—the ’Fractional Preferential Attachment’ (FPA) scale-free network model—that generates networks with time-independent degree distributions $p\left(k\right)∼k^{-\gamma }$ with degree exponent 2<γ≤3 (where γ=3 corresponds to the typical value of the BA model). In the FPA model, the element controlling the network properties is the *f* parameter, where f∈(0,1〉. Depending on the different values of *f* parameter, we study the statistical properties of the numerically generated networks. We investigate the topological properties of FPA networks such as degree distribution, degree correlation (network assortativity), clustering coefficient, average node degree, network diameter, average shortest path length and features of fractality. We compare the obtained values with the results for various synthetic and real-world networks. It is found that, depending on *f*, the FPA model generates networks with parameters similar to the real-world networks. Furthermore, it is shown that *f* parameter has a significant impact on, among others, degree distribution and degree correlation of generated networks. Therefore, the FPA scale-free network model can be an interesting alternative to existing network models. In addition, it turns out that, regardless of the value of *f*, FPA networks are not fractal.

## 1. Introduction and Motivation

The last three decades of multidisciplinary research have shown that networks are ubiquitous in various complex systems of nature and society. Scale-free networks are of particular importance. This type of networks can describe diverse systems as communication systems: the Internet [1,2,3] and the www [4,5,6]; social systems, e.g., food web [7], the web of human sexual contacts [8], language [9], the web of actors [10] or scientific-collaboration networks [11,12,13]; biological systems, e.g., the cell (a network of substrates connected by chemical reactions), the protein networks [14,15]; and the structure of the mountain ridge [16].

The simplest and one of the most important characteristic of a single node in a network is its degree (k = 0, 1, 2, 3, ... N). The range of node degrees over a network is characterised by a distribution function p(k), which is the probability that a randomly selected node has *k* edges. The networks whose degree distribution fulfil a power-law:$$p\left(k\right)=Ck^{-\gamma }$$
are called ’scale-free networks’. Furthermore, if a network is directed, the scale-free property can apply independently to the in- and the out-degrees. Power-law distributions are more general than an exponential one [17]—they do not have a characteristic scale and allow very large nodes to exist.

As listed in Table 1 in [18,19], the numerical values of degree exponent γ for different systems are various and depend on the details of network structure, but most of them appear to have γ between 2 and 3 (or 1 and 2 for cumulative degree distributions P(k)). From the viewpoint of science, it would be interesting to find a universal network model that would generate scale-free networks with any parameter γ. A few sample network models and numerical results are presented in [18] (Table 1). These models are generally based on constant network growth and the preferential addition of links to nodes.

Numerical results for real networks have indicated that, in comparison with random graphs [17] having the same size and the same average degree, the average path length of the scale-free network is somewhat smaller; nonetheless, the clustering coefficient is much higher. This means that, in scale-free networks, there are relatively few nodes with large degree (’hub’) that play a key role in bringing other nodes closer to each other. It is known that real networks are open and dynamically formed by continuous addition of new nodes to the network. The network models presented in the literature are often static in the sense that, although edges can be added or rearranged, the number of nodes is determined throughout the forming process. The preferential attachment is generally understood as a mechanism where newly arriving nodes have a tendency to connect with already well-connected nodes [12,20,21]. The concept of preferential attachment has received great attention in broadly understood world science and has been used to explain many observations in a variety of real-world networks. For example, social scientists started to use this concept in order to explain social processes—among others, the explanation of the concept of evolving co-authorship networks [22,23], investigation of collaboration networks in science and IT industry and to find that preferential attachment acts as a main mechanism in the evolution of these networks [12,24]. The concept of preferential attachment refers to the observation that, in networks with growth over time, the probability an edges is added to a node with *k* neighbours is proportional to *k*. This is a typical linear relationship where the number of nodes with *k* is proportional to $k^{-\gamma }$. Models with non-linear preferential attachment of nodes have also been proposed; however, they generally do not lead to a power-law of degree distribution in the whole range of *k* [25].

The Barabási–Albert (BA) scale-free network model [26] is the best known and recognised theoretical model that takes into account the coexistence of growth and preferential attachment. The model is defined as follows. We start with $m_{0}$ nodes, the links between which are chosen arbitrarily, as long as each node has at least one link. At each time-step, we add a new node with *m* links that connect the new node to *m* nodes of the network. The probability P(i) that a link of the new node connects to node *i* depends on the degree $k_{s}$ as:(1)$$P\left(i\right)=\frac{k_{i}}{\sum_{j}k_{j}}.$$

The classical version of the BA model assumes γ=3 [26], while in the case of real networks often γ≠3. In addition, the mechanism of linear preferential attachment in the BA model determines the topological parameters of the network, which often differ from the parameters of real networks. In this article, we propose a generalisation of the model by adding parameter *f*, which allows generating scale-free networks with any scaling parameter 2<γ≤3. In addition, all network models described in the literature assume that during the network formation process all nodes previously present in the network are ’seen’. We propose a model where the mechanism of preferential attachment of the next node does not take into account all existing nodes but only a part of them. In this way, we weaken the rule of linear preferential attachment in a global sense.

## 2. The Fractional Preferential Attachment Scale-Free Network Model

### 2.1. The Model

One of the most known preferential attachment models that produce scale-free networks is the BA model [26], in which the initial degree of each new node is *m*. However, networks generated by this model have two properties that often differ from the properties of many real networks. The first one is related to the adopted scheme of the node ordering, in which a given parent node (i.e., the node with an order *R*) often has an equal or larger number of branches than any of its neighbours (R+1). It obviously implies that degrees of the corresponding nodes remain in the same relation: $k^{\left(R\right)}\geq k^{\left(R+1\right)}$. However, this relation may not be valid in the network created according to the BA model—in this model, we define the parent nodes as the older ones in each pair of linked nodes. Such a situation can be observed if there is a node with a smaller degree than degrees of its two or more neighbours (see Figure 1).

The second difference is that the BA model creates networks with the CDF (β=γ−1) scaling exponent β=2, while in the case of real networks often β≠2. The most widely used models that are able to simulate growth of the scale-free networks with β<2, such as the Dorogovtsev–Mendes–Samukhin model [27], require allowing cycles to exist. Therefore, we propose a different approach.

We introduce two modifications to the original BA algorithm in order to obtain an agreement with the empirical networks. We preserve its basic feature, i.e., the preferential attachment. In each step *j*, a new node *j* is linked to the already existing network by a single edge (m=1), whose opposite end is to be attached to some candidate node *i* chosen with the probability proportional to this node’s degree $k_{i}$. However, before this link may be established, degrees $k_{i}$ and $k_{p(i)}$ are compared, where p(i) denotes the parent node of the candidate node *i*. If $k_{i}\leq k_{p(i)}$, the link is established between *i* and *j* and the node *i* is attributed with the parenthood for *j*: i=p(j), otherwise the link connects *j* with p(i) and p(i)=p(j). Topology of the so-modified network excludes now the situation mentioned above (see Figure 1b), while the degree distribution remains the same as in the original BA model.

The second modification is based on restricting the set of existing nodes that a new node *j* can be linked to. This is done by ordering the j−1 nodes according to their degree from the highest to the lowest one and selecting only a subset F of $n_{j-1}$ nodes that belong to a given upper fraction $f=n_{j-1}/\left(j-1\right)$ (0<f≤1). If two or more nodes have the same degree, we order them according to the order *R* (lower *R*s first) and, in the case of the same *R*s, the order of such nodes is random. For f=1, all the existing nodes are taken into consideration like in the BA model. If f<1, the preferential attachment mechanism is restricted to the subset F where not all nodes are considered as candidates to which another node can join—if *f* decreases, the number of candidates also decreases. The probability *P* of drawing the node *i* is given by
(2)$$P\left(i\right)=\frac{k_{i}}{\sum_{s=1}^{n_{j-1}}k_{s}}.$$

Note that, if a node belongs to F, the same is true for its parent node. Figure 2 shows a sample network with the distinguished nodes belonging to F. We call this model “The Fractional Preferential Attachment” (FPA) scale-free network model henceforth.

### 2.2. the Network Topology

The networks generated by the FPA model are shown in Figure 3.

In all cases, the network generated by the model has N=40,000 nodes (to clearly visualise the network, the largest possible number of nodes was selected—a larger number of nodes causes details to be unreadable).

The visualisation is presented for six different values of 0<f≤1. We can observe that, for decreasing values of *f*, the network has an increasingly extensive single node (’hub’). On the other hand, for f=1, the network becomes a BA network. In our considerations, we show examples of FPA networks for relatively small values of *m* because such networks are most common in nature. An example would be networks with m=1, where the attached node connects to only one node in the existing network. The real networks with m=1 take the form of acyclic networks containing open cycles, i.e., paths that start and end at the same node and follow edges only in the ahead direction. The best known examples are citation networks but there are many others as well, such as family trees, phylogenetic networks, food webs, feed-forward neural networks and call graphs. In addition, we can often successfully approach the cyclic networks with the acyclic one.

First, we study the degree distributions of FPA network model. Figure 4 exhibits the CDF $P\left(k\right)∼k^{-\beta }$ for different values of *f* for m=1 and m=2. In all the cases, the network generated by the model has N=100,000 nodes.

It is clearly seen that the networks generated with the FPA model have the scale-free degree distribution with *f*-dependent power law exponent β. For f=1, the exponent β≈2.0, as expected for the generic BA model and this value is the largest one. By decreasing the *f* parameter, we observe that the P(k) degree distribution becomes fatter and β decreases up to β=1.1 for f=0.1. This means that FPA model can generate a scale-free networks with any exponent from the range 1<β≤2. Figure 4 shows the results only for the case m=1 and m=2. Table 2 shows β values also for m=3,5. It turns out that, as in the classical BA model, the values of power-law exponent β do not change when the *m* parameter increases. In addition, the results of numerical simulations show that the value of the scaling parameter β is closely related to *f*, i.e., β=f+1 and PDF scaling exponent γ=β+1=f+2 has the values 2<γ≤3. Many real-world networks have the nature of scale-free networks, where the scaling parameter of degree distributions has values that the FPA model can generate. Documented examples of this type of networks are shown in Table 1.

Other basic network parameters generated by the FPA model depending on the *f* parameter are shown in Table 2.

Presented values were obtained as the average of 10 independent realisations, each with N=100,000 nodes. The analysis has been carried out for different *m* values. This parameter defines the number of network nodes to which the linked (new) node connects. The results are shown for m={1,2,3,5} and for f={0.1,0.3,0.5,0.7,0.9,1}. The average local clustering coefficient $\varrho =\sum_{k}C_{j}/N$ of a network is the average of $C_{j}$ over all the nodes, where *N* is the number of nodes and $C_{j}=2x/\left(k_{j}\left(k_{j}-1\right)\right)$ is the local clustering coefficient of node *i* [33]. $C_{j}$ is the ratio of the total number *x* of the links connecting its nearest neighbours to the total number of all possible links between all these nearest neighbours, where $k_{j}$ is the degree of node *j*. The ϱ values are shown in Table 2. It is seen that, for m=1, regardless of the *f* parameter, the value ϱ=0. Interestingly, we observe ϱ>0 for m>1, where the maximum value of clustering coefficient ϱ occurs at f=0.1, i.e., when the network has the most numerous hubs. Then, as *f* increases, the value of ϱ decreases up to 0 for f=1 (typical for classical BA model). Furthermore, the average degree 〈k〉 and the network diameter *D* for different *f* are shown in Table 2. Regardless of the *f* parameter, the values of 〈k〉 agree the relationship typical for the classic BA model, i.e., 〈k〉≈2m. However, we observe the dependence of the *f* parameter on the network diameter *D*—if *D* decreases, *f* decreases as well. This effect is caused by the fact that when *f* decreases, the attraction force of subsequent nodes to nodes with a high degree (hubs) increases, which in turn causes degeneration of network growth.

Another important property of the networks generated by FPA model refers to the relation between the average shortest path length *L* and the number of nodes *N*. This relationship fulfils the conditions typical for ultra-small world network, i.e., ln(N)/ln(ln(N))≈L≈4.7 [34]. What is more, this relation is fulfilled not only for m≥2 and f=1 (classical BA model) as proved in the literature [34] but also for m=1 (the graph turns into trees) when f<0.5. It is noted that this condition is even more fulfilled for m≥2 and f<1. Furthermore, it should be clearly emphasised that the small-networks condition mentioned above is true only for uncorrelated networks [34]. To quantify this statement, we examine the correlation level of FPA network nodes. The degree correlations measures the relationship between the degrees of nodes that link to each other. One way to quantify their magnitude is measuring for each node *i* the average degree of its neighbours: $k_{nn}\left(k_{i}\right)=1/k_{i}\sum_{j=1}^{N}K_{ij}k_{j}$. The degree correlation function calculated for all nodes with degree *k* is given by [3,35]:(3)$$\langle k_{nn}\left(k\right)\rangle =\sum_{k^{{}^{\prime }}}k^{{}^{\prime }}P\left(k^{{}^{\prime }}|k\right),$$
where $P(k^{{}^{\prime }}|k)$ is the conditional probability that following a link of a *k* degree node we get a $k^{{}^{\prime }}$ degree node. Therefore, $\langle k_{nn}\left(k\right)\rangle$ is the average degree of the neighbours of all *k*-degree nodes. To quantify degree correlations, we analyse the possible power-law dependence of $k_{nn}$ on *k*. If the relation
(4)$$\langle k_{nn}\left(k\right)\rangle ∼k^{\alpha }$$
holds, the magnitude and the nature of degree correlations is determined by the correlation exponent α. When α is positive (α>0), the network is assortative; if α=0, the degree correlation is neutral; and, if α<0, the graph is disassortative.

The results for different *f* parameters and for m=1 are shown in Figure 5. It is clear that degree correlation function (Equation 3) (the network assortativity) strictly depends on *f*—the α assortativity parameter changes smoothly from 0 to −1. We observe the highest dissortativity for f=0.1 (α=−1). Then, when *f* increases, the dissortativity (i.e., the degree of power correlation) decreases and is close to 0 for the classical BA network (f=1). The values of the α scaling parameter for m={1,2,3,5} are shown in Table 2. It is seen that a similar nature of degree correlation is also observed for higher *m* values. However, the dissortativity decreases more slowly with increasing *f*. The α=0 value obtained here for f=1 is consistent with current knowledge, while the appearance of a relatively high level of degree correlation for smaller *f* is an interesting observation especially in the context of real network modelling. Interestingly, many scale free networks of practical interest, from the WWW or e-mail to citation, *E. coli* metabolism or protein interaction networks, have a dissortative nature [35,36,37]. Therefore, by selecting the appropriate parameter *f* for the FPA model, we can generate network with a dissortative correlation of nodes degree consistent with the observed real network.

### 2.3. the Fractal Analysis

To identify possible fractal nature of networks generated by the FPA model, we selected the renormalisation-based approach method, which is the most adequate method of quantifying network fractality, in this case the box-covering method [38]. In the box-covering method, a network is divided into node clusters (boxes) and subsequently coarse-grained on various ‘length’ scales. One first defines the distance parameter *r* that bounds the path length between the nodes belonging to the same cluster. Next, a seed node that is considered as a centre of the first cluster is randomly chosen. Then, the number $Q_{c}$ of the clusters is calculated after partitioning the network into the node clusters in such a way that the minimum path length between the nodes belonging to the same cluster is not longer than r−1. The same is done independently for different draws of the seed node and the average number of clusters $\langle Q_{c}\rangle$ is determined. In the next step, the renormalisation, the clusters are replaced by single nodes (linked if there existed at least one connection between the nodes belonging to the corresponding clusters) and a new network consisting of such nodes is formed. These steps can be repeated until the network is reduced to a single cluster. However, since the consecutive renormalised networks have similar topological properties to the original network, we calculate $\langle Q_{c}\left(r\right)\rangle$ for different values of the parameter *r*. The network is fractal if the following relation is fulfil:(5)$$\langle Q_{c}\left(r\right)\rangle /N∼r^{-D_{f}},$$
where $\langle Q_{c}\rangle$ is the average number of the network clusters, *N* is the number of the network nodes and *r* is the distance parameter. Furthermore, the $D_{f}$ parameter is considered to be the fractal dimension of the network. It is documented in the literature that the empirical networks can show either the fractal scaling (Equation (Equation 5)) or a non-fractal behaviour of $\langle N_{c}\left(l\right)\rangle /N$, e.g., its exponential decay [38]. We use the box-covering procedure for the networks generated by FPA model for six parameters of *f*. For each *f*, the network has 100,000 nodes (the same networks as in Table 2). The results for m={1,3,5} are shown in Figure 6. It is clear that, despite the fact that these networks are scale-free, they are non-fractal, i.e., do not present the power-law dependence (Equation (Equation 5)). This situation is observed regardless of the selected *f* parameter. Instead, the exponential relation $\langle Q_{c}\left(r\right)\rangle ≃ae^{-r/b}$, (*a* and *b* are constants) where $\langle Q_{c}\rangle$ is the average number of the network clusters, approximate well the empirical data for all the values of *l* in this case. The main reason for the lack of multifractal character of FPA networks may be just their scale-free nature with power-law distributions and 2<γ≤3. In this regime of γ, the first moment of the degree distribution is finite but the second and higher moments tend to infinity. Consequently, scale-free networks in this regime are ultra-small or even smaller, as shown above. From a different perspective, the scale-free character of networks generated by the FPA model implies the existence of massive hubs, and this, by forming highly populated node clusters, limits the total number of clusters in the network even for the moderate values of *r* and produces the exponential decay of $\langle Q_{c}\left(r\right)\rangle$ [16,38,39].

## 3. Summary and Conclusions

The last two decades of research on complex networks have led to a huge development of this field of science. The language of complex networks allowed to see large similarities in the topology between such seemingly diverse systems as the Internet, the electronic circuits, language and even financial systems. Today’s level of technology development allows for an empirical analysis of large-scale and complex dynamical networks generated by nature, as well as enables the construction and verification of more sophisticated models. This in turn contributes to better understanding of the processes in the world around us as well as to a more accurate prediction of their future behaviors (states).

In summary, we propose a network model which is a generalisation of the well-known Barabási–Albert model. The main feature of both models is the preferential attachment, which in general is a very natural property (rich get richer, Matthew effect). The advantage of the FPA model compared to the BA model is that the FPA model has parameter *f* by which we can control the power of the preferential attachment. This power is inversely proportional to the f parameter—if *f* decreases, the power of preferential attachment increases. We can look at the *f* parameter from a different perspective. As opposed to the BA model, our model assumes that in the process of adding subsequent nodes not all existing nodes are included in the preferential attachment statistics. The process of the degree of ’invisibility’ is controlled by the parameter 0<f≤1. For example, if f=0.7, then the mechanism of preferential attachment considers 70% of all existing nodes (f=0.7) and therefore does not take into account the 30% of nodes with the smallest degree. In the case the degree of nodes is the same, the oldest nodes enter the 30% invisible nodes. The validity of this approach can be imagined on the example of the WWW—at a given moment some servers are not active in a specific time zone but still belong to the network.

Both models can generate scale-free networks. The classical version of the BA model generate scale-free networks with only one γ=3 scaling parameter. The presented FPA model can generate scale-free networks with the scaling degree distribution parameter in theoretically the most interesting regime 2<γ≤3. This is undoubtedly a huge advantage of the FPA model because this range of γ values is characteristic for real-world networks. Importantly, numerical studies have shown a close relation between *f* and γ: γ=f+2. If f=1, then the classic BA model is realised. On the other hand, if f→0, then γ→2. An interesting property of the FPA model from the point of view of real networks is that there are no big clusters (hubs) on the periphery of the network, i.e., there are no connections between nodes of high degree through the node of low degree. An example is the network of air connections where connections between large cities do not take place through small airports. However, such links may appear in the BA model. Another curious result of our study worth further investigation is that FPA networks have dissortative nature despite the fact that they are small-world type. We have also shown that, if *f* increases, the dissortativity (degree correlation) decreases and is close to 0 for the classical BA network (for f=1).

Of course, the presence of the *f* parameter in our model makes it more complicated in terms of computations and programming. However, these costs are small compared to the advantages of the FPA model. We have also shown that FPA networks, similar to BA networks, are not fractal. In certain circumstances, this can be considered as a disadvantage of our model. However, we have an idea how to obtain the fractal (or maybe even multifractal) networks and at the same time keep their scale-free character. In the FPA model presented here, we assume that, in the process of creating a network, the value of *f* parameter is constant (for BA f=1). This might be the reason we get non-fractal networks. Therefore, the next stage of our study is the FPA model, where the value of *f* will change in each next step of adding a new node. It is possible to assume various variants of choosing the *f* value—random samples from different distribution (e.g., Gauss, triangular or uniform) when the series is not fractal or random samples from fractal or even multifractal series (with long-term correlations).

To summarise, quantitative analyses of FPA networks have revealed their interesting topological properties, where, with proper selection of the *f* parameter, they can (better than the classical BA model) reflect the features of real networks.

## Figures and Tables

**Figure 1 entropy-22-00509-f001:**
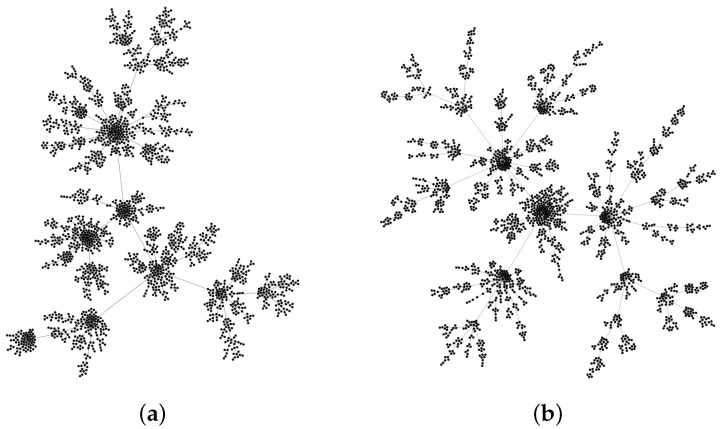
The networks simulated with: (**a**) the generic Barabási–Albert model (m=1, i.e., a new node is connected by only one edge); and (**b**) the modified model, in which an older node in any pair of the linked nodes cannot be less connected than the newer one.

**Figure 2 entropy-22-00509-f002:**
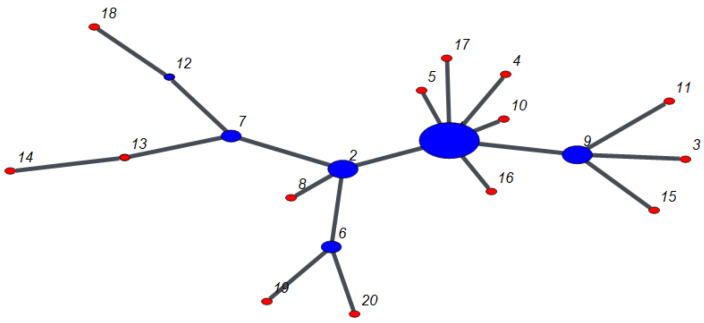
A snapshot of a network growing according to the FPA scale-free model discussed in this work. Only the fraction f=0.3 of the most connected nodes (blue) will be considered for preferential attachment in the next step, while the remaining ones (red) will be left as they are. The network is acyclic, i.e., m=1 (tree network).

**Figure 3 entropy-22-00509-f003:**
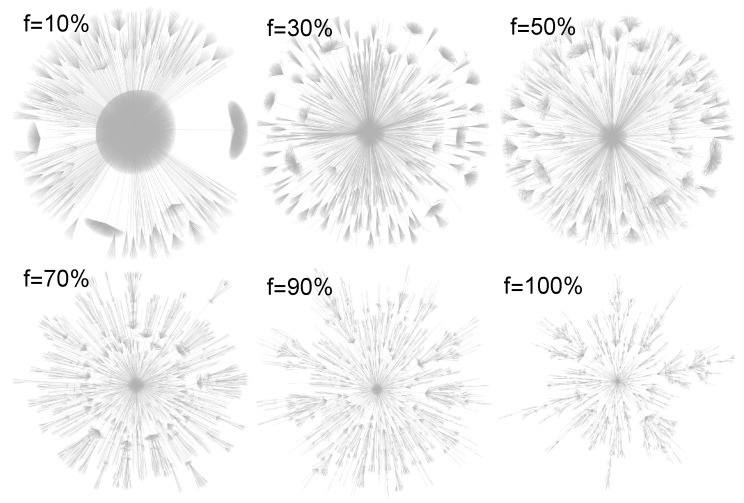
The structure of networks generated by FPA scale-free network model for different values of parameter *f*. In all cases, N=40,000 nodes. All networks are acyclic, i.e., m=1 (tree networks).

**Figure 4 entropy-22-00509-f004:**
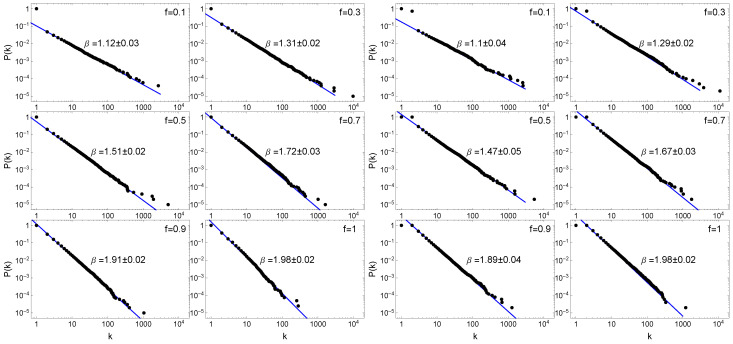
Log-log plot of the cumulative degree distributions for generated by FPA model for different values of 0<f≤1. Two cases are shown: for acyclic (tree) networks where m=1 (**left**); and for networks with m=2 (**right**). The blue solid lines correspond to the best theoretical fits of the function $∼k^{-\beta }$. All values β were averaged over 10 independent realisations, each with N=100,000 nodes. The error (±) denotes the standard deviation derived from 10 independent realisations.

**Figure 5 entropy-22-00509-f005:**
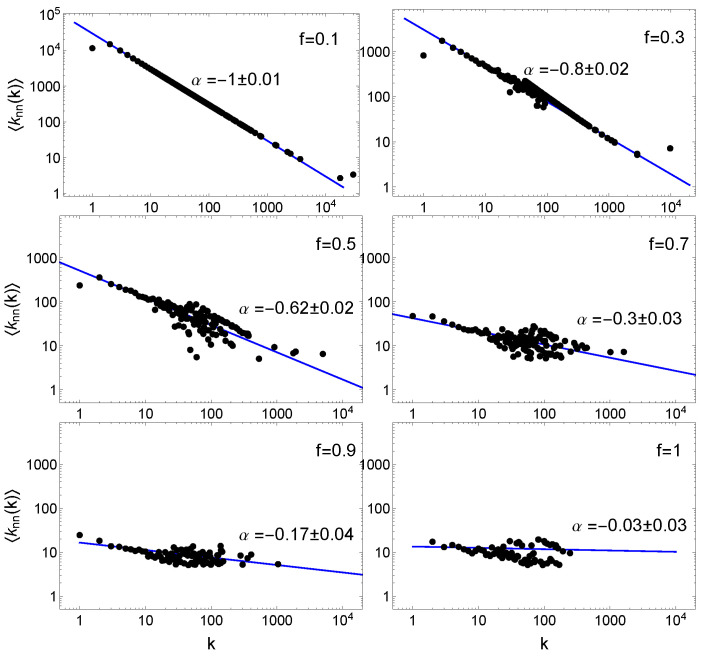
Log-log plot of the degree correlation function $\langle k_{nn}\left(k\right)\rangle$ for acyclic networks (m=1) generated by FPA model for different values 0<f≤1. The blue solid lines correspond to the best theoretical fits of the function $∼k^{-\alpha }$. The results are presented for the same networks as in Figure 4. The error (±) denotes the standard deviation derived from 10 independent realisations.

**Figure 6 entropy-22-00509-f006:**
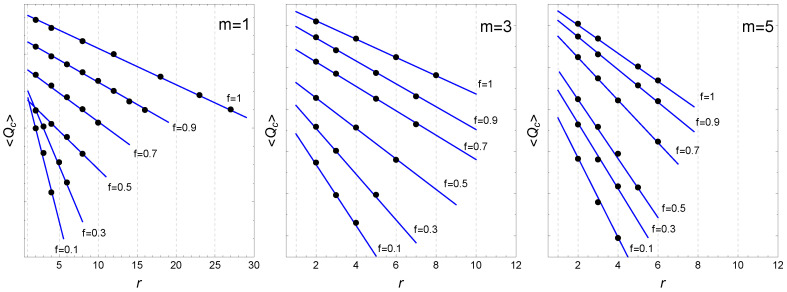
The average number of clusters $\langle Q_{c}\left(r\right)\rangle$ for the same networks as in Table 2. The exponential function g(r)=aexp(−r/b) is shown by the straight blue lines on semi-log scale. The graphs for the individual *f* are shifted vertically in relation to each other for better visualisation.

**Table 1 entropy-22-00509-t001:** Scaling parameter of PDF degree distributions for real-world networks.

Scale-Free Network	γ	Ref.
World Wide Web link networks	2.1–2.7	[4]
Internet connections at the router level	2.4–2.5	[1]
Actor cooccurrence in films	2.3	[26,28]
Scientific collaboration networks	2.5–3	[11,13]
Mountain ridge networks	2.6–2.7	[16]
Scientific paper citation networks	3	[29]
Word-cooccurrence networks	2.8	[30]
Protein interaction networks	2.4	[15]
Biochemical cellular pathway	2–2.4	[14]
Currency comovement networks	2.4–2.7	[31,32]

**Table 2 entropy-22-00509-t002:** The basic topological parameters of FPA model for m={1,2,3,5} depending on parameter f: the correlation exponent α (assortativity), the average local clustering coefficient ϱ, the average node degree 〈k〉, the network diameter *D*, the average shortest path length *L*, the maximum node degree $k_{\max}$ and the power-law scaling exponent β. All values were averaged over 10 independent realisations, each with N=100,000 nodes. The error (±) denotes the standard deviation derived from 10 independent realisations.

f	α	ϱ	〈k〉	*D*	*L*	$k_{\max}$	β
*m* = 1							
0.1	−1±0.01	0	2	4	3.3	29,819	1.12±0.03
0.3	−0.80±0.02	0	2	6	4.4	9726	1.31±0.02
0.5	−0.62±0.03	0	2	8	5.4	4981	1.51±0.02
0.7	−0.30±0.03	0	2	10	7.2	1621	1.72±0.03
0.9	−0.17±0.02	0	2	16	9.5	1038	1.91±0.02
1	−0.03±0.01	0	2	33	11.6	825	1.98±0.02
*m* = 2							
0.1	−0.99±0.02	0.48	3.3	4	2.5	39,849	1.1±0.04
0.3	−0.92±0.03	0.11	3.9	6	3.1	15,050	1.29±0.02
0.5	−0.59±0.02	0.03	4	6	4	7395	1.47±0.05
0.7	−0.4±0.03	0	4.1	8	4.8	3771	1.67±0.03
0.9	−0.24±0.02	0	4.1	9	5.2	1873	1.89±0.04
1	−0.12±0.03	0	4.2	10	5.5	1207	1.98±0.02
*m* = 3							
0.1	−0.99±0.02	0.61	5.6	4	2.19	58,123	1.13±0.03
0.3	−0.91±0.02	0.13	5.9	5	3.2	26,823	1.3±0.02
0.5	−0.64±0.02	0.04	6	6	3.6	16,125	1.51±0.02
0.7	−0.48±0.02	0.01	6	7	4.1	7020	1.69±0.02
0.9	−0.27±0.03	0.003	6	7	4.5	3589	1.89±0.02
1	−0.16±0.02	0	6	8	4.7	2852	2.02±0.03
*m* = 5							
0.1	−0.96±0.01	0.58	9.1	4	2.05	75,652	1.09±0.02
0.3	−0.90±0.02	0.17	9.6	4	2.81	42,873	1.31±0.02
0.5	−0.78±0.02	0.06	9.9	5	3.19	33,129	1.48±0.03
0.7	−0.46±0.02	0.013	9.9	6	3.6	10,435	1.71±0.02
0.9	−0.38±0.02	0	10.1	6	3.78	6141	1.89±0.02
1	−0.21±0.02	0	10.2	6	3.99	3948	1.98±0.03

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
