# Peer review of "The Fractional Preferential Attachment Scale-Free Network Model"

_entropy, 2020, doi:10.3390/e22050509_

Round 1

Reviewer 1 Report

In this paper a new model to generate scale-free networks is proposed:
the Fractional Preferential Attachment model (FPA). This model is
based on preferential attachment algorithm, but it considers two basic
properties: (I) An adopted scheme of the node ordering, in which a
given parent node (i.e., the node with an order R) has often an equal
or larger number of branching than any of its neighbors. II) The
second difference is that the BA model creates networks with the CDF
(\beta=\gamma-1) scaling exponent \beta = 2, while in the case of real
networks \beta != 2 very often. The most used models to do so require
allowing cycles to exist.

The authors introduce two modifications to the Barabasi-Albert (BA)
model of preferential attachment. i) They conserve the preferential
attachment but, they compare attached nodes to guarantee condition
(I). ii)The second modification is based on restricting the set of
existing nodes that a new node can be linked to. It is achieved
by considering a fraction parameter f that controls to which nodes the
new nodes may be attached.

After presenting the model, different properties of its model are
shown and compared with real networks, arguing that the FPA reproduces
real network properties in a more accurate manner than the preferential
attachment model. Some measures of comparison are the correlation exponent
(assortativity), the average local clustering coefficient, the average
node degree, the network diameter, the average shortest path length,
the maximum node degree, and the power-law scaling exponent. Finally,
it turned out that regardless of the value of f, FPA networks are not
fractal.

Major Comment:

It is interesting to note that a simple mechanism (but not simpler
than the original BA model) allows to produce networks with properties
that can be found in real networks. However, it deserves a major
discussion about the computational and calculation advantages of
considering FPA. The authors should describe in detail the advantages and
disadvantages of the model in order to consider a more complex
algorithm to generate scale-free networks. In this respect, the
authors should present if the computational costs deserve to be
assumed against considering the BA model or other variants.

Minor comments:

The paper is easy to read and well written.

Line 67-68: a new idea is introduced with "-" but closed with a ")".

Once these comments are resolved, this work may be considered to be published in the journal.

Author Response

We are grateful to Referees' for their remarks, which we found very insightful.

All the changes, including minor ones, have been marked in red in the attached paper.

Below please find our response to the to reviewers' comments.

Reviewer 2 Report

Models allow to reproduce most important features of real systems, discarding non necessary elements and retaining only essential ones.
This is also true for network models, which try and mimic real networks.
One of the most successful network models has been proposed by Barabasi and Albert (BA), as it reproduces the scale-free nature of real-world networks. This means that the higher moments (>2) of the degree distribution are infinite, thus allowing for the existence of nodes exhibiting a huge number of connections.
The BA model creates a network using two mechanisms: the addition of nodes and the preferential attachment. The first mimics the dynamic nature of networks, while the second describes that new nodes do not connect to randomly chosen nodes already belonging to the networks. Rather, new nodes tend to connect to nodes having the largest number of connections. It is well known that both mechanisms are necessary to successfully reproduce the scale-free character of real networks.
In this manuscript an extension of the BA model is presented in which: i) the degree of a node cannot exceed the degree of the parent node and ii) the number of nodes a new node can connect to depends on a parameter f.
These two modifications allow to create a rich and interesting scenario of situations ranging from scale-free characteristic of model networks to (dis)assortativity and fractality.
The contribution is really interesting and I believe it may represent a step further of the state-of-the-art knowledge of model networks.
Some minor changes may help in improving the quality of the paper:
1) introduce the main features of BA model in the Introduction, together with eq. (1).
2) summarize values of $\gamma$ in lines 115-125 in a table
3) avoid referring to column number in lines 131 and 135-136. Instead, use the column heading
4) Given that experiments with m=1,3,5 clearly describe main features of the newly introduced model, an experiment with m=2 or 4 may help. Do you expect something different when m is even? I do not.
5) in line 33 authors claim that "in scale-free networks, there are several large nodes with very large degrees ('hubs')". The power-law degree distribution is a clear evidence that large(?) nodes with very large degrees are rare. What are large nodes? Why there several large nodes?
6) In Section 2.2 (lines 95-100) authors claim that the choice of m=1 is corroborated by 'many real networks'. While this is true, a lot of real networks are more closely described by m=2 or m=3. Why they haven't added something similar to Figure 4 with m=2 or 3?
Here and there the manuscript needs proofread. For example:
lines 2 and 3 'preferential attachment nodes' -> preferential attachment
lines 24 and 28: Table1 -> Table 1
line 45: 'the probability of an edge is added...' -> the probability an edges is added
line 99: software callgraphs -> call graphs.

Author Response

(The authors gave the same response as above.)

Round 2

Reviewer 1 Report

The authors have answered all the comments made. So I suggest to accept.